# An Iteratively Extended Target Tracking by Using Decorrelated Unbiased Conversion of Nonlinear Measurements

**DOI:** 10.3390/s24051362

**Published:** 2024-02-20

**Authors:** Yuemei Qin, Yang Han, Shuying Li, Jun Li

**Affiliations:** School of Automation, Xi’an University of Posts & Telecommunications, Xi’an 710121, China; hanyang@stu.xupt.edu.cn (Y.H.); lishuying@xupt.edu.cn (S.L.); 1923239208@stu.xupt.edu.cn (J.L.)

**Keywords:** extended target tracking, nonlinear measurements, decorrelated unbiased technique, iterative estimation

## Abstract

Extended target tracking (ETT) based on random matrices typically assumes that the measurement model is linear. However, nonlinear measurements (such as range and azimuth) depending on locations of a series of unknown scattering centers always exist in many practical tracking applications. To address this issue, this paper proposes an iteratively extended target tracking based on random matrices by using decorrelated unbiased conversion of nonlinear measurements (ETT-IDUCM). First, we utilize a decorrelated unbiased converted measurement (DUCM) method to convert nonlinear measurements depending on unknown scatters of target extent in polar coordinates into the ones in Cartesian coordinates with equivalent measurement noise covariances. Subsequently, a novel method, combining iteratively extended Kalman filter (IEKF) updates with variational Bayesian (VB) cycles is developed for precise estimation of the target’s kinematic state and extension. This method leverages the synergy between external IEKF iterations, which use the estimated state as a new prediction and input for DUCM, and internal VB iterations, which realize a closed-form approximation of the joint posterior probability. This approach progressively enhances estimation accuracy. Simulation results demonstrate the ETT-IDUCM algorithm’s superior precision in estimating the target’s kinematic state and extension compared to existing methods.

## 1. Introduction

Target tracking aims to estimate the kinematic state of moving targets using acquired measurements [1]. In traditional radar tracking, targets are often considered as point sources of measurement due to sensor resolution constraints and significant measurement errors relative to the target size [2]. This implies that the tracking process primarily estimates the target’s kinematic state, while neglecting extensions such as the target’s size and shape [3]. However, as the resolution of modern sensors (such as phased array radar) improves, it becomes possible to observe multiple measurements from a single target in each scan [4,5]. Under these circumstances, treating targets as point targets becomes inadequate, due to the need for detection, classification, and recognition of various types of targets [3,6]. Moreover, modern radar devices can extract detailed target information, which is beneficial for identification and tracking, based on multiple measurements. Consequently, a target might be considered as an extended target (ET), characterized by parameters such as its kinematic state, size, shape, direction, and so on. The number of measurements produced by such extended targets can vary, and measurement uncertainty can arise due to factors such as target extension and the geometric relationship between the sensor and the target [6]. Additionally, associating the results of a single measurement with the scattering center presents a significant challenge. In these complex situations, a practical approach is to consider group targets as a single entity. By sidestepping the issues of kinematic estimation and data association for individual targets, extended target tracking methods can be effectively utilized for group target tracking problems.

The existing literature provides numerous models for individual extended target tracking (ETT), such as the spatial distribution model [7], the random hyper-surface model [8], the Gaussian process model for star-convex shapes [9], and the multiplicative error model [10], among others. The random matrix approach was first proposed in [3]. It estimates the random vector representing the kinematic state of the target’s centroid and the symmetric positive definite (SPD) random matrix representing the target’s extension based on the measurements. This method estimates the state and extension of the tracking target without using measurement association methods, and its final computation process is much simpler compared to other methods, making it quite promising for application. Within the random matrix framework, many further developments have been proposed, including [11,12,13,14,15]. Among them, Lan and Li [12,13], respectively, use two reversible matrices to describe complex dynamic changes and specific extension evolution models. The aforementioned random matrix models generally rely on a forgetting factor to handle statistical data of unknown extension matrices to interpret changes in target orientation, or use information obtained from target trajectories to estimate the target’s orientation, which leads to poorer tracking performance when the target is maneuvering. Therefore, a new method for ETT was proposed in [14], which can simultaneously estimate the kinematic, extension, and orientation states of an extended target, accurately modeling the changes in the orientation of the extended target.

Existing random matrix approaches typically assume that measurements are linear in the kinematic state and measurement noise. However, in many practical tracking systems, measurements exhibit nonlinearity, depending on the location of the scattering center. Given this, to apply random matrix approaches to nonlinear measurements, linearization must first be considered. There are two main strategies for addressing linearization issues. The first strategy involves converting measurements between polar/spherical coordinates and Cartesian coordinates, followed by estimating the covariance of the converted measurement error. The conventional form of conversion uses a first-order Taylor expansion to obtain the covariance after conversion [16]. However, this conventional form of conversion has an inherent drawback: related methods may cause a significant reduction in numerical accuracy for highly nonlinear measurement functions. Another unbiased conversion method can produce a biased estimate in the Bayesian recursion because the covariance after conversion is correlated with the measurement error [17]. Both forms of conversion require two prior assumptions. If these assumptions do not hold, numerical accuracy may decrease. The second strategy involves the linearization of the measurement function. In [12], Lan and Li proposed a general random matrix method for extended target tracking (ETT) using a linearization function. In fact, this method cannot only obtain a linear state form but also retain the second moment of the extension information. However, it requires the inversion of the coefficient matrix to ensure the implementation of linearization. Unfortunately, in practice, when the matrix becomes singular, the inversion cannot always be satisfied, especially noting that matrix inversion may cause the filter to fail.

In fact, without considering extended targets, there are many other effective methods for handling nonlinear measurements. For instance, adaptive and robust Kalman filtering based on the variational Bayesian method [18,19] not only estimates unknown statistical information about system noise but also suppresses outliers in measurements. It adaptively estimates the noise covariance for unknown or time-varying measurements. Additionally, the deep Kalman filters method [20,21] integrates the principles of deep learning with Kalman filtering by training neural networks to learn the system’s dynamic and observation models, employing nonlinear approximation schemes, such as deep neural networks, to manage nonlinearity. Similarly, the deep reinforcement learning method [22] also merges deep learning with Kalman filtering principles, using neural networks to learn the system’s dynamics and observation models, and applying nonlinear approximation techniques to address nonlinearity.

In summary, recent years have seen a lack of particularly effective processing methods for extended target tracking scenarios based on radar data, which provide measurements with radial distance and orientation. Existing various linearization models are not applicable in this scenario. However, in radar and sonar systems, radial distance and orientation are the most widely used, which is the focus of this article. Additionally, in many emerging fields, it is more appropriate to describe the target state in polar coordinates. Therefore, to address these issues, this article proposes a new random matrix method for ETT using nonlinear measurements.

To make the random matrix method effective, we consider using the first strategy to complete the linearization process of nonlinear measurements. The conversion methods based on Kalman filtering that have been proposed so far mainly include traditional trigonometric transformations, unbiased converted measurement (UCM) [17], modified UCM (MUCM) [23], and unscented transform (UT) [24]. These four conversion methods are affected by different degrees of estimation bias, due to the covariance after conversion being related to sensor error. This correlation can lead to biased estimation in Bayesian recursion. To solve this problem, Bordonaro and others proposed a decorrelated UCM (DUCM) technique to overcome conversion and estimation bias [25], obtaining the conversion covariance of the predicted kinematic state to avoid the impact of correlation. Therefore, we use DUCM for measurement linearization, apply it to the ETT proposed in [14], and perform cyclic estimation in the state update part to improve the accuracy and robustness of state estimation, and better handle nonlinearity and dynamic changes.

This paper makes the following contributions:(1)The decorrelated unbiased conversion measurement (DUCM) technique transforms nonlinear measurements from polar to Cartesian coordinates, enabling random matrix-based tracking of nonlinear measurements. This method ensures high-precision conversion, laying the groundwork for accurate kinematic state and extension estimation.(2)We introduce a novel approach by integrating the iterated extended Kalman filter (IEKF) with the variational Bayesian (VB) method, employing random matrices for updating the target’s kinematic state and extension. This integration not only improves estimation accuracy but also introduces a new direction for future research. The combination of IEKF and VB iterations refines the estimation process, leading to precise estimates of the target’s state and extension.

Notations: For clarity, we use italics to represent scalars and boldface italics to represent vectors and matrices. We use ‘:=’ to define a quantity, and $A^{T}$ to represent the transpose of a vector or matrix A. The n-dimensional identity matrix is represented by $I_{n}$.

The structure of this paper is as follows: Section 2 states a measurement model for ETT based on nonlinear measurements. Section 3 proposes an ETT method using VB-based IEKF. Section 4 presents a simulation comparison between the proposed model and a series of existing ones. Finally, Section 5 provides a summary of the entire paper.

## 2. ETT with Nonlinear Measurements in Polar Coordinates and Its Measurement Conversion

### 2.1. ETT Using Random Matrix

The use of a random matrix to describe the extended state of ETT was first proposed in the literature by [3]. It assumes that the estimated state consists of the kinematic state xk∈ℝnx and the symmetric positive definite (SPD) target extension matrix consists of Xk∈ℝny×ny, where nx and ny represent the dimensions of the kinematic state in one-dimensional space and physical space, respectively. Let Zk:={zkj}j=1nk be the set of nk independent Cartesian position measurements within the scan at time k. In [3], the measurement model is as follows:(1)zkj=Hkxk+vkj, j=1,…,nk
where Hk is the measurement matrix, vkj~N(0,sXk+Rk) is the measurement noise, the scalar s is used to describe the scaling factor of the extension state, and Rk is the sensor error covariance.

In the study by [14], a new measurement model was proposed that can separate the uncertainty of the orientation parameter and shape parameter into two additive Gaussian terms:(2)$$z_{k}^{j}=H_{k}x_{k}+v_{k}^{I}\left(X_{k},\theta_{k}\right)+v_{k}^{\mathrm{II}},\;j=1,\ldots ,n_{k}$$
where $v_{k}^{I}\left(X_{k},\theta_{k}\right)\sim N\left(0,sT_{\theta_{k}}X_{k}T_{\theta_{k}}^{T}\right)\;$ has time-varying, unknown, but extent state-dependent statistics, and $v_{k}^{\mathrm{II}}\sim N(0,R_{k})$ has known statistical data, where $x_{k}$ is the target kinematic state, $X_{k}$ is the target extent matrix, $\theta_{k}$ is the orientation angle, $T_{\theta_{k}}$ is the rotation matrix, and $T_{\theta_{k}}≜\left[\begin{array}{cc} \cos\left(\theta_{k}\right) & -\sin\left(\theta_{k}\right)\\ \sin\left(\theta_{k}\right) & \cos\left(\theta_{k}\right) \end{array}\right]$.

### 2.2. Nonlinear Measurements Conversion from Polar to Cartesian Coordinates with Covariance

For data obtained from radar systems, measurements of the target position are typically provided in polar coordinates (i.e., range and azimuth). However, target motion is usually modeled in Cartesian coordinates. Therefore, traditional linear Kalman filters can only be applied after the measurement data has been converted from polar coordinates to Cartesian coordinates. For tracking results, it is crucial to properly consider the impact of this conversion.

For the extended target, consider the jth measurement $\zeta_{k}^{j}$ in the polar coordinate, where its measurement model at time k is described as follows:(3)$$\zeta_{k}^{j}:=\left[\begin{array}{c} r_{k}^{j}\\ \varepsilon_{k}^{j} \end{array}\right]={\left[\begin{array}{c} \sqrt{{\left(x_{k}+u_{x_{k}}^{j}\right)}^{2}+{\left(y_{k}+u_{y_{k}}^{j}\right)}^{2}}+v_{r_{k}}^{j}\\ \arctan\left(\frac{y_{k}+u_{y_{k}}^{j}}{x_{k}+u_{x_{k}}^{j}}\right)+v_{\varepsilon_{k}}^{j} \end{array}\right]}_{k}$$
where $r_{k}^{j}$ and $\varepsilon_{k}^{j}$ are, respectively, the range and azimuth radar measurements at time k, ${\left[x_{k},y_{k}\right]}^{T}$ is the target centroid position, ${\left[u_{x_{k}}^{j},u_{y_{k}}^{j}\right]}^{T}$ is the distance from the radar scattering point to the centroid, and ${\left[v_{r_{k}}^{j},v_{\varepsilon_{k}}^{j}\right]}^{T}$ is the range and azimuth measurement noise.

Let us define the variables $x_{k}^{\mathrm{pos}}:={\left[x,y\right]}_{k}^{T}$, $v_{k}^{I}:={\left[u_{x}^{j},u_{y}^{j}\right]}_{k}^{T}$ and $v_{k}^{II}:={\left[v_{r}^{j},v_{\varepsilon }^{j}\right]}_{k}^{T}$, where $x_{k}^{\mathrm{pos}}=H_{k}x_{k}$ is the centroid location of the ET, $v_{k}^{I}\sim N\left(0,sT_{\theta_{k}}X_{k}T_{\theta_{k}}^{T}\right)\;$ denotes the measurement data generated near the centroid of the ET, and their range can be characterized by $sT_{\theta_{k}}X_{k}T_{\theta_{k}}^{T}$. $v_{k}^{\mathrm{II}}\sim N(0,R_{k})$ represents the sensor errors, following a zero-mean Gaussian distribution with standard deviations $\sigma_{r}$ and $\sigma_{\varepsilon }$. Here, $x_{k}^{\mathrm{pos}}$, $v_{k}^{I}$, and $v_{k}^{\mathrm{II}}$ are mutually independent.

When applying the random matrix method for modeling, the conversion of nonlinear measurements is first considered. The advantage of this method is that it can overcome both conversion bias and estimation bias. To convert measurements from polar coordinates to Cartesian coordinates, at time k, we use the decorrelated unbiased conversion (DUCM):(4)$$z_{k}^{L,j}≜{\left[x_{k}^{L,j},y_{k}^{L,j}\right]}^{T}={\left[e^{\sigma_{\varepsilon }^{2}/2}r_{k}^{j}\cos\varepsilon_{k},e^{\sigma_{\varepsilon }^{2}/2}r_{k}^{j}\sin\varepsilon_{k}\right]}^{T}$$
where ${\left[x_{k}^{L,j}:=x_{k}+u_{x_{k}}^{j},y_{k}^{L,j}:=y_{k}+u_{y_{k}}^{j}\right]}^{T}$ represents the location of the scattering point on the extended target’s contour (rather than the location of the geometric center of the contour).

Assume that at time *k*, there is a set of $n_{k}$ independent linearized measurements, as mentioned in the literature [3,6], denoted as $Z_{k}:={\left\{z_{k}^{L,j}\right\}}_{j=1}^{n_{k}}$. Therefore, the likelihood function of the measurement set is given as:(5)$$p\left(Z_{k}|n_{k},x_{k},X_{k},\theta_{k}\right)=\prod_{j=1}^{n_{k}}p\left(z_{k}^{L,j}|x_{k},X_{k},\theta_{k}\right)$$

Each measurement $z_{k}^{L,j}$ is modeled as a noisy measurement of a noiseless point $y_{k}^{j}$, located somewhere on the extended target. The likelihood function of individual measurements is as follows:(6)$$p\left(z_{k}^{L,j}|x_{k},X_{k},\theta_{k}\right)=\int p\left(z_{k}^{L,j}|y_{k}^{j},x_{k},X_{k},\theta_{k}\right)\times p\left(y_{k}^{j}|x_{k},X_{k},\theta_{k}\right)dy_{k}^{j}$$

Note that we can obtain $p\left(z_{k}^{L,j}|x_{k},X_{k},\theta_{k}\right)$ by marginalizing $y_{k}^{j}$.

For the type of radar system considered here, the measurement noise is accurately modeled as a zero-mean Gaussian type, so based on (6) we can get:(7)$$p\left(z_{k}^{L,j}|y_{k}^{j},x_{k},X_{k},\theta_{k}\right)=N\left(z_{k}^{L,j};y_{k}^{j},R\left(y_{k|k-1}^{j,p}\right)\right)$$
where ${\hat{y}}_{k|k-1}^{j}:={\left[x_{t},y_{t}\right]}^{T}$ represents the position prediction of $y_{k}^{j}$, and $R\left({\overset{\;}{y}}_{k|k-1}^{j}\right)$ is the covariance estimate of the coordinate conversion. Let $r_{t}:=\sqrt{{\left(x_{t}\right)}^{2}+{\left(y_{t}\right)}^{2}}$ and $\varepsilon_{t}:=\arctan\left(y_{t}/x_{t}\right)$ represent the predicted distance and orientation, with $\sigma_{r_{t}}^{2}$ and $\sigma_{\varepsilon_{t}}^{2}$ being the corresponding variances, i.e.,
(8)$$\begin{array}{l} \sigma_{r_{t}}^{2}=\frac{p_{\mathrm{xx}}x_{t}^{2}+2p_{\mathrm{xy}}x_{t}y_{t}+p_{\mathrm{yy}}y_{t}^{2}}{x_{t}^{2}+y_{t}^{2}}\\ \sigma_{\varepsilon_{t}}^{2}=\frac{p_{\mathrm{xx}}y_{t}^{2}-2p_{\mathrm{xy}}x_{t}y_{t}+p_{\mathrm{yy}}x_{t}^{2}}{{\left(x_{t}^{2}+y_{t}^{2}\right)}^{2}} \end{array}$$
where $\left[\begin{array}{ll} p_{\mathrm{xx}} & p_{\mathrm{xy}}\\ p_{\mathrm{yx}} & p_{\mathrm{yy}} \end{array}\right]$ is the covariance of the predicted state ${\hat{y}}_{k|k-1}^{j}$.

Then, we can get the converted covariance estimation $R\left({\hat{y}}_{k|k-1}^{j}\right)$ as follows:(9)$$R\left({\hat{y}}_{k|k-1}^{j}\right)=\left[\begin{array}{ll} R_{11}\left({\hat{y}}_{k|k-1}^{j}\right) & R_{12}\left({\hat{y}}_{k|k-1}^{j}\right)\\ R_{21}\left({\hat{y}}_{k|k-1}^{j}\right) & R_{22}\left({\hat{y}}_{k|k-1}^{j}\right) \end{array}\right]$$
where the elements can be defined as follows:(10a)$$R_{11}\left({\hat{y}}_{k|k-1}^{j}\right)=\frac{1}{2}\left(r_{t}^{2}+\sigma_{r}^{2}+\sigma_{r_{t}}^{2}\right)\times \left[1+\cos\left(2\varepsilon_{t}\right)e^{-2\sigma_{\varepsilon }^{2}}e^{-2\sigma_{\varepsilon_{t}}^{2}}\right]e^{\sigma_{\varepsilon }^{2}}-\frac{1}{2}\left(r_{t}^{2}+\sigma_{r_{t}}^{2}\right)\left[1+\cos\left(2\varepsilon_{t}\right)e^{-2\sigma_{\varepsilon_{t}}^{2}}\right]$$
(10b)$$R_{22}\left({\hat{y}}_{k|k-1}^{j}\right)=\frac{1}{2}\left(r_{t}^{2}+\sigma_{r}^{2}+\sigma_{r_{t}}^{2}\right)\times \left[1-\cos\left(2\varepsilon_{t}\right)e^{-2\sigma_{\varepsilon }^{2}}e^{-2\sigma_{\varepsilon_{t}}^{2}}\right]e^{\sigma_{\varepsilon }^{2}}-\frac{1}{2}\left(r_{t}^{2}+\sigma_{r_{t}}^{2}\right)\left[1-\cos\left(2\varepsilon_{t}\right)e^{-2\sigma_{\varepsilon_{t}}^{2}}\right]$$
(10c)$$R_{12}\left({\hat{y}}_{k|k-1}^{j}\right)=\frac{1}{2}\left(r_{t}^{2}+\sigma_{r}^{2}+\sigma_{r_{t}}^{2}\right)\times \left[\sin\left(2\varepsilon_{t}\right)e^{-2\sigma_{\varepsilon }^{2}}e^{-2\sigma_{\varepsilon_{t}}^{2}}\right]e^{\sigma_{\varepsilon }^{2}}-\frac{1}{2}\left(r_{t}^{2}+\sigma_{r_{t}}^{2}\right)\left[\sin\left(2\varepsilon_{t}\right)e^{-2\sigma_{\varepsilon_{t}}^{2}}\right]$$

According to the literature [14], for elliptical targets, the scattering center $z_{k}^{j}$ is approximated as the following Gaussian distribution:(11)$$p\left(y_{k}^{j}|x_{k},X_{k},\theta_{k}\right)=N\left(y_{k}^{j};H_{k}x_{k},sT_{\theta_{k}}X_{k}T_{\theta_{k}}^{T}\right)$$

Combined with (6), the likelihood function is the marginalization of $y_{k}^{j}$, that is,
(12)$$\begin{array}{ll} p\left(z_{k}^{L,j}|x_{k},X_{k},\theta_{k}\right) & =\int N\left(z_{k}^{L,j};y_{k}^{j},R\left({\hat{y}}_{k|k-1}^{j}\right)\right)\times N\left(y_{k}^{j};H_{k}x_{k},sT_{\theta_{k}}X_{k}T_{\theta_{k}}^{T}\right)dy_{k}^{j}\\ & =N\left(z_{k}^{L,j};H_{k}x_{k},sT_{\theta_{k}}X_{k}T_{\theta_{k}}^{T}+R\left({\hat{y}}_{k|k-1}^{j}\right)\right) \end{array}$$

Since it is hardly possible to accurately infer the reflection point $y_{k}^{j}$ on the target contour from the radar plot $z_{k}^{j}$, it becomes unfeasible to provide an accurate position prediction ${\hat{y}}_{k|k-1}^{j}$ and the corresponding prediction error covariance $R\left({\hat{y}}_{k|k-1}^{j}\right)$ for each reflection point.

Therefore, the position prediction of the extended target’s geometric center ${\hat{x}}_{k|k-1}$ is used as substitutes for each scattering point. Its prediction covariance is denoted as $\hat{R}$.

Through the above calculations, we finally obtain the measurement model of (2) in the Cartesian coordinate system:(13)$$z_{k}^{L,j}=H_{k}x_{k}+v_{k}^{j}$$
where $v_{k}^{j}\sim N\left(0,sT_{\theta_{k}}X_{k}T_{\theta_{k}}^{T}+\hat{R}\right)$.

**Remark** **1.***When*$\hat{R}$ *is a constant matrix, this approximation is easy to hold. Generally, when *$\hat{R}$ *does not change significantly within the uncertain area of the extended target, this assumption approximately holds. However, when the change in the covariance of the measurement noise is too large, the performance of the related estimator will decrease, or even diverge. Considering this impact, the proposed linearization model can only be realized when it is assumed that the target kinematic state *$x_{k}$ *, extension state *$X_{k}$ *, and orientation angle *$\theta_{k}$ *are independent of each other. Under the above assumptions, compared with the method proposed in [12], the linearized measurement model avoids the operation of matrix inversion, and therefore has better numerical stability.*

## 3. Variational Bayesian-Based Iteratively Extended Kalman Filtering for ETT

Before introducing Section 3 in detail, we will first define the prior distribution of unknown variables. The joint prior distribution of the target’s kinematic state, range, and direction is specified as follows:(14)$$p\left(X_{0},X_{0},\theta_{0}\right):=N\left(x_{0};{\hat{x}}_{0},P_{0}\right)\times \prod_{i=1}^{n_{y}}ℐG\left(\sigma_{0}^{i};\alpha_{0}^{i},\beta_{0}^{i}\right)\times N\left(\theta_{0};{\hat{\theta }}_{0},\Theta_{0}\right)$$
where, $X_{0}≜\mathrm{diag}(\sigma_{0}^{1},\sigma_{0}^{2},\ldots ,\sigma_{0}^{n_{y}})$, $ℐG\left(\sigma_{0}^{i};\alpha_{0}^{i},\beta_{0}^{i}\right)$ represents the inverse Gamma distribution, ${\hat{x}}_{0}$ and $P_{0}$ are the prior mean and covariance matrix of the Gaussian motion state vector, respectively. The prior mean and covariance matrix of the orientation angle $\theta_{0}$ are ${\hat{\theta }}_{0}$ and $\Theta_{0}$, respectively.

### 3.1. Time Update

Under the random matrix framework, we can assume that the dynamic models of the motion state and extension state are independent,
(15)$$p\left(x_{k},X_{k}|x_{k-1},X_{k-1}\right)=p\left(x_{k}|x_{k-1}\right)p\left(X_{k}|X_{k-1}\right)$$

The time update of the motion state follows the Kalman filter prediction equation:(16)$$x_{k}^{a}=Fx_{k-1}^{a}+u_{k},u_{k}\sim N\left(0,Q\right)$$
where $x_{k}^{a}≜{\left[x_{k}^{T},\theta_{k}\right]}^{T}$.

The predicted posterior density $N\left(x_{k|k-1}^{a};{\hat{x}}_{k|k-1}^{a},P_{k|k-1}^{a}\right)$ is obtained by updating the mean and covariance of the Gaussian component according to system dynamics:(17)$${\hat{x}}_{k|k-1}^{a}=F{\hat{x}}_{k-1|k-1}^{a}$$
(18)$$P_{k|k-1}^{a}=FP_{k-1|k-1}^{a}F^{T}+Q$$
where $P_{k}^{a}≜\mathrm{blkdiag}\left(P_{k},\Theta_{k}\right)$.

In most target tracking, the exact dynamics of the extension state are unknown. If it is slowly varying but unknown, the forgetting factor can be used to predict the target extension state. The parameter update of the inverse Gamma distribution can be expressed as the following equation:(19)$$\alpha_{k|k-1}^{i}=\gamma \alpha_{k-1|k-1}^{i}$$
(20)$$\beta_{k|k-1}^{i}=\gamma \beta_{k-1|k-1}^{i},\;\mathrm{for}\;i=1,\ldots ,n_{y}$$
where γ is the forgetting factor.

### 3.2. Measurement Update

Before we introduce the measurement update, we need to define additional instrumental variables to solve the conjugacy loss caused by the additive measurement noise covariance term $\hat{R}$ in the likelihood function. We call these instrumental variables noiseless measurements [26] and represent them with $Y_{k}:={\left\{y_{k}^{j}\right\}}_{j=1}^{n_{k}}$. Using $Y_{k}$, in the variational approximation calculation of the algorithm we proposed, the joint posterior density about the target motion state, orientation angle, and extension state can be approximately represented as the product of the following densities:(21)$$p\left(x_{k},X_{k},\theta_{k},Y_{k}|Z_{1:k}\right):=q_{x}\left(x_{k}\right)q_{X}\left(X_{k}\right)q_{\theta }\left(\theta_{k}\right)q_{Y}\left(Y_{k}\right)$$

Using the Bayesian calculation rule, the joint density $p\left(x_{k},X_{k},\theta_{k},Y_{k},Z_{k}|Z_{1:k-1}\right)$ can be rewritten explicitly as follows:(22)$$\begin{array}{ll} p\left(x_{k},X_{k},\theta_{k},Y_{k},Z_{k}|Z_{1:k-1}\right)= & p\left(Z_{k}|Y_{k}\right)p\left(Y_{k}|x_{k},X_{k},\theta_{k}\right)p\left(x_{k},X_{k},\theta_{k}|Z_{1:k-1}\right)\\ & =\left(\prod_{j=1}^{n_{k}}N\left(z_{k}^{j};y_{k}^{j},\hat{R}\right)\right)\left(\prod_{j=1}^{n_{k}}N\left(y_{k}^{j};H_{k}x_{k},sT_{\theta_{k}}X_{k}T_{\theta_{k}}^{T}\right)\right)\\ & \times N\left(x_{k};{\hat{x}}_{k|k-1},P_{k|k-1}\right)\prod_{i=1}^{n_{y}}ℐG\left(\sigma_{k|k-1}^{i};\alpha_{k|k-1}^{i},\beta_{k|k-1}^{i}\right)\times N\left(\theta_{k};{\hat{\theta }}_{k|k-1},\Theta_{k|k-1}\right) \end{array}$$

The Equation (22) can be solved by fixed-point iteration. In each iteration, an approximate probability is calculated, and the approximate density of the unknown parameters is obtained respectively.

Since the equivalent measurement noise covariance from DUCM depends on the one-step predicted target location ${\hat{x}}_{k}$, we consider the iterated extended Kalman filter (IEKF) method here. That is, we replace the estimated value of the target state and its corresponding covariance with the new results obtained from the last variational Bayesian cycle, and then perform the iteration of EKF. In each iteration, we will repeat the above update process. We denote the number of iterations in the state update part of IEKF as the subscript n. Therefore, using Equation (22), we can get the update result of the (ℓ+1)th variational cycle during the (n+1)th filter update. Building on the derivation approaches in references [14,26], and based on the innovation filtering, the following measurement update can be obtained:▪Update the kinematic state:
(23)$$q_{x_{n+1}}^{\left(\ell +1\right)}\left(x_{k}\right)=N\left(x_{k};{\hat{x}}_{n+1,k|k}^{\left(\ell +1\right)},P_{n+1,k|k}^{\left(\ell +1\right)}\right)$$
where
(24a)$${\hat{x}}_{n+1,k|k}^{\left(\ell +1\right)}=P_{n+1,k|k}^{\left(\ell +1\right)}(P_{n,k|k-1}^{-1}{\hat{x}}_{n,k|k-1}+n_{k}H_{k}^{T}{\tilde{\Lambda }}_{k}{\bar{y}}_{k}))$$
(24b)$$P_{n+1,k|k}^{\left(\ell +1\right)}={\left(P_{n,k|k-1}^{-1}+n_{k}H_{k}^{T}{\tilde{\Lambda }}_{k}H_{k}\right)}^{-1}$$
with ${\bar{y}}_{k}:=\frac{1}{n_{k}}\sum_{j=1}^{n_{k}}{\hat{y}}_{n,k}^{j,\left(\ell \right)}$,
(24c)$${\tilde{\Lambda }}_{k}:=E_{q_{X_{n}}^{\left(\ell \right)},q_{\theta_{n}}^{\left(\ell \right)}}\left[{\left(sT_{\theta_{k}}X_{k}T_{\theta_{k}}^{T}\right)}^{-1}\right]=\left(1-\exp\left(-2\Theta_{n,k|k}^{\left(\ell \right)}\right)\right)\frac{\mathrm{Tr}{\tilde{\Delta }}_{k}}{2}I_{2}+\exp\left(-2\Theta_{n,k|k}^{\left(\ell \right)}\right)\left(T_{{\hat{\theta }}_{n,k|k}^{\left(\ell \right)}}\tilde{\Delta }T_{{\hat{\theta }}_{n,k|k}^{\left(\ell \right)}}^{T}\right)$$
(24d)$${\tilde{\Delta }}_{k}:=E_{q_{X_{n}}^{\left(\ell \right)}}\left[{\left(sX_{k}\right)}^{-1}\right]=\mathrm{diag}\left(\frac{\alpha_{n}^{1,\left(\ell \right)}}{s\beta_{n}^{1,\left(\ell \right)}},\frac{\alpha_{n}^{2,\left(\ell \right)}}{s\beta_{n}^{2,\left(\ell \right)}},\ldots ,\frac{\alpha_{n}^{n_{y},\left(\ell \right)}}{s\beta_{n}^{n_{y},\left(\ell \right)}}\right)$$

▪Update the extension state:

(25)$$q_{X_{n+1}}^{\left(\ell +1\right)}\left(X_{k}\right)=\prod_{i=1}^{n_{y}}ℐG\left(\sigma_{n+1,k|k}^{i,\left(\ell +1\right)};\alpha_{n+1,k|k}^{i,\left(\ell +1\right)},\beta_{n+1,k|k}^{i,\left(\ell +1\right)}\right)$$
where
(26a)$$\alpha_{n+1,k|k}^{i,\left(\ell +1\right)}=\alpha_{n,k|k-1}^{i}+0.5n_{k}$$
(26b)$$\beta_{n+1,k|k}^{i,\left(\ell +1\right)}=\beta_{n,k|k-1}^{i}+\frac{1}{2s}\sum_{j=1}^{n_{k}}{\tilde{\Psi }}_{k}$$
with
(26c)$${\tilde{\Psi }}_{k}:=E_{q_{\theta_{n}}^{\left(\ell \right)}}\left\{T_{\theta_{n,k}^{\left(\ell \right)}}^{T}\left[\left({\hat{y}}_{n,k}^{j,\left(\ell \right)}-H_{k}{\hat{x}}_{n,k|k}^{\left(\ell \right)}\right){\left({\hat{y}}_{n,k}^{j,(n)}-H_{k}{\hat{x}}_{n,k|k}^{\left(\ell \right)}\right)}^{T}+H_{k}P_{n,k|k}^{\left(\ell \right)}H_{k}^{T}+\Sigma_{n,k}^{y,\left(\ell \right)}\right]T_{\theta_{n,k}^{\left(\ell \right)}}\right\}$$

According to Lemma 1 in [14], we know the following:(27a)$$E_{q_{\theta_{n}}^{\left(\ell \right)}}{\left[{\left(T_{\theta_{n,k}^{\left(\ell \right)}}MT_{\theta_{n,k}^{\left(\ell \right)}}^{T}\right)}^{-1}\right]}_{11}=\left[m_{11}m_{22}-\left(m_{12}+m_{21}\right)\right]K\left({\hat{\theta }}_{n,k|k}^{\left(\ell \right)},\Theta_{n,k|k}^{\left(\ell \right)}\right)$$
(27b)$$E_{q_{\theta_{n}}^{\left(\ell \right)}}{\left[{\left(T_{\theta_{n,k}^{\left(\ell \right)}}MT_{\theta_{n,k}^{\left(\ell \right)}}^{T}\right)}^{-1}\right]}_{12}=\left[m_{12}-m_{21}m_{11}-m_{22}\right]K\left({\hat{\theta }}_{n,k|k}^{\left(\ell \right)},\Theta_{n,k|k}^{\left(\ell \right)}\right)$$
(27c)$$E_{q_{\theta_{n}}^{\left(\ell \right)}}{\left[{\left(T_{\theta_{n,k}^{\left(\ell \right)}}MT_{\theta_{n,k}^{\left(\ell \right)}}^{T}\right)}^{-1}\right]}_{21}=\left[m_{21}-m_{12}m_{11}-m_{22}\right]K\left({\hat{\theta }}_{n,k|k}^{\left(\ell \right)},\Theta_{n,k|k}^{\left(\ell \right)}\right)$$
(27d)$$E_{q_{\theta_{n}}^{\left(\ell \right)}}{\left[{\left(T_{\theta_{n,k}^{\left(\ell \right)}}MT_{\theta_{n,k}^{\left(\ell \right)}}^{T}\right)}^{-1}\right]}_{22}=\left[m_{22}m_{11}m_{12}+m_{21}\right]K\left({\hat{\theta }}_{n,k|k}^{\left(\ell \right)},\Theta_{n,k|k}^{\left(\ell \right)}\right)$$
where
(27e)$$K\left({\hat{\theta }}_{n,k|k}^{\left(\ell \right)},\Theta_{n,k|k}^{\left(\ell \right)}\right)≜\left[\begin{array}{c} 1+\cos\left(2{\hat{\theta }}_{n,k|k}^{\left(n\right)}\right)\exp\left(-2\Theta_{n,k|k}^{\left(n\right)}\right)\\ 1-\cos\left(2{\hat{\theta }}_{n,k|k}^{\left(n\right)}\right)\exp\left(-2\Theta_{n,k|k}^{\left(n\right)}\right)\\ \sin\left(2{\hat{\theta }}_{n,k|k}^{\left(n\right)}\right)\exp\left(-2\Theta_{n,k|k}^{\left(n\right)}\right) \end{array}\right]$$
with $M^{-1}=\left[\begin{array}{cc} m_{11} & m_{12}\\ m_{21} & m_{22} \end{array}\right]$.

Substituting (26c) into (27) can yield the corresponding expectations.

▪Update the tool variable:

(28)$$q_{y_{n+1}}^{\left(\ell +1\right)}\left(y_{k}\right)=\prod_{i=1}^{n_{k}}N\left(y_{k};{\hat{y}}_{n+1,k|k}^{\left(\ell +1\right)},\Sigma_{n+1,k|k}^{\left(\ell +1\right)}\right)$$
where
(29a)$${\hat{y}}_{n+1,k}^{j,\left(\ell +1\right)}=\Sigma_{n+1,k}^{y,\left(\ell +1\right)}({\tilde{\Lambda }}_{k}H{\hat{x}}_{n,k|k}^{\left(\ell \right)}+{{\hat{R}}_{n+1}}^{-1}z_{k}^{j})$$
(29b)$$\Sigma_{n+1,k}^{y,\left(\ell +1\right)}={\left({\tilde{\Lambda }}_{k}+{{\tilde{R}}_{n+1}}^{-1}\right)}^{-1}$$
with ${\hat{R}}_{n+1}:=R\left({\hat{x}}_{n+1,k|k-1}^{\left(\ell +1\right)}\right)$.

**Remark** **2.***The role of introducing the instrumental variable *$Y_{k}$ *is just to derive the posterior density of the target state one by one. Therefore, it is only calculated in the process of the variational Bayesian cycle, and the initial value *$y_{k}^{j,\left(0\right)}:=z_{k}^{j}$ *and covariance *$\Sigma_{k}^{y,\left(0\right)}$ *remain unchanged:*


(30)
$$\Sigma_{k}^{y,\left(0\right)}:=E_{q_{X_{n}}^{\left(\ell \right)}}\left[\left(sX_{k}\right)\right]=\mathrm{diag}\left(\frac{s\beta_{n}^{1,\left(0\right)}}{\left(\alpha_{n}^{1,\left(0\right)}-1\right)},\frac{s\beta_{n}^{2,\left(0\right)}}{\left(\alpha_{n}^{2,\left(0\right)}-1\right)},\ldots ,\frac{s\beta_{n}^{n_{y},\left(0\right)}}{\left(\alpha_{n}^{n_{y},\left(0\right)}-1\right)}\right)$$


▪Update the orientation angle:

(31)$$q_{\theta_{n+1}}^{\left(\ell +1\right)}\left(\theta_{k}\right)=N\left(\theta_{k};{\hat{\theta }}_{n+1,k|k}^{\left(\ell +1\right)},\Theta_{n+1,k|k}^{\left(\ell +1\right)}\right)$$
where
(32a)$${\hat{\theta }}_{n+1,k|k}^{\left(\ell +1\right)}=\Theta_{n+1,k|k}^{\left(\ell +1\right)}\left\{\Theta_{n,k|k-1}^{-1}{\hat{\theta }}_{n,k|k-1}+{\tilde{A}}_{k}\right\}$$
(32b)$$\Theta_{n+1,k|k}^{\left(\ell +1\right)}={\left\{\Theta_{n,k|k-1}^{-1}+{\tilde{B}}_{k}\right\}}^{-1}$$
with
(32c)$${\tilde{A}}_{k}:=\sum_{j=1}^{n_{k}}\mathrm{Tr}\left[{\tilde{\Delta }}_{k}{\left(T_{{\hat{\theta }}_{n,k|k}^{\left(\ell \right)}}^{\prime }\right)}^{T}{\tilde{\Omega }}_{k}\left(T_{{\hat{\theta }}_{n,k|k}^{\left(\ell \right)}}^{\prime }\right){\hat{\theta }}_{n,k|k}^{\left(\ell \right)}\right]-\mathrm{Tr}\left[\tilde{\Delta }T_{{\tilde{\Lambda }}_{n,k|k}^{\ell }}^{T}{\tilde{\Omega }}_{k}\left(T_{{\hat{\theta }}_{n,k|k}^{\left(L\right)}}^{\prime }\right)\right]$$
(32d)$${\tilde{B}}_{k}:=\sum_{j=1}^{n_{k}}\mathrm{Tr}\left[\tilde{\Delta }{\left(T_{{\hat{\theta }}_{n,k|k}^{\left(\ell \right)}}^{\prime }\right)}^{T}{\tilde{\Omega }}_{k}\left(T_{{\hat{\theta }}_{n,k|k}^{\left(\ell \right)}}^{\prime }\right)\right]$$
(32e)$${\tilde{\Omega }}_{k}:=E_{q_{Y_{n}}^{\left(\ell \right)},q_{x_{n}}^{\left(\ell \right)}}\left[\left(y_{k}^{j}-H_{k}x_{k}\right){\left(y_{k}^{j}-H_{k}x_{k}\right)}^{T}\right]=H_{k}P_{n,k|k}^{\left(\ell \right)}H_{k}^{T}+\Sigma_{n,k}^{y,\left(\ell \right)}+\left({\hat{y}}_{n,k}^{j,\left(\ell \right)}-H_{k}{\hat{x}}_{n,k|k}^{\left(\ell \right)}\right){\left({\hat{y}}_{n,k}^{j,\left(\ell \right)}-H_{k}{\hat{x}}_{n,k|k}^{\left(\ell \right)}\right)}^{T}$$
(32f)$$\;{T_{{\hat{\theta }}_{n,k|k}^{\left(\ell \right)}}^{\prime }≜\frac{\partial T_{\theta_{k}}}{\partial \theta_{k}}|}_{\theta_{k}={\hat{\theta }}_{n,k|k}^{\left(\ell \right)}}$$

Due to the nonlinearities involved in $E_{q_{x_{n}}^{\left(\ell \right)}q_{X_{n}}^{\left(\ell \right)},q_{\theta_{n}}^{\left(\ell \right)},q_{Y_{n}}^{\left(\ell \right)}}\left[\mathrm{Tr}\left[\left(y_{k}^{j}-H_{k}x_{k}\right){\left(y_{k}^{j}-H_{k}x_{k}\right)}^{T}\times {\left(sT_{\theta_{k}}X_{k}T_{\theta_{k}}^{T}\right)}^{-1}\right]\right]$, to attain an exact compact form of the PDF for $q_{\theta_{n+1}}^{\left(\ell +1\right)}\left(\theta_{k}\right)$, a Taylor series expansion around ${\hat{\theta }}_{n,k|k}^{\left(\ell \right)}$ is employed to approximate the nonlinear function $f(\theta_{k})≜T_{\theta_{k}}^{T}\left(y_{k}^{j}-H_{k}x_{k}\right)$. Through this approximation, the variables ${\tilde{A}}_{k}$ and ${\tilde{B}}_{k}$ are derived.

A summary of the resulting iterative measurement update procedure is given in Algorithm 1. Table 1 discusses the computational complexity [27] of one iteration of the proposed ETT-IDUCM process by providing the number of flops for each step. Here, xk∈ℝn,αk,βk∈ℝm and yk∈ℝm×p.

## 4. Simulation and Discussion

In this section, we will compare the extended target tracking method for nonlinear measurements proposed above with a method proposed in [6]. The algorithm, which utilizes standard coordinate conversion, is denoted as Fel-SCM. To intuitively demonstrate the effectiveness of the proposed algorithm, we also chose the ETT methods proposed in [5,14] with single mode, both of which utilize UCM technology for coordinate conversion. These are denoted as Tun-UCM and Wen-UCM, respectively.

### 4.1. Comparison of Tracking Accuracy of Different Algorithms

In the simulation, we use the root mean square error (RMSE) of the target centroid position, motion speed, area, and the Gaussian–Wasserstein distance (GWD) mentioned in [28] as parameters to evaluate and compare the performance of the algorithm.

GWD includes the position error and shape error of the target:
(33)$$d{\left(\mu_{1},X_{1},\mu_{2},X_{2}\right)}^{2}:={\left\|\mu_{1}-\mu_{2}\right\|}^{2}+\mathrm{Tr}\left\{X_{1}+X_{2}-2\sqrt{\sqrt{X_{1}}X_{2}\sqrt{X_{1}}}\right\}$$

**Algorithm 1** Variational measurement update.Input: $\left\{{\hat{x}}_{k|k-1},P_{k|k-1}\right\},{\left\{\alpha_{k|k-1}^{i},\beta_{k|k-1}^{i}\right\}}_{i=1}^{n_{y}},\left\{{\hat{\theta }}_{k|k-1},\Theta_{k|k-1}\right\},Z_{k}$
Output: ${\hat{x}}_{k|k},P_{k|k},{\left\{\alpha_{k|k}^{i},\beta_{k|k}^{i}\right\}}_{i=1}^{n_{y}},{\hat{\theta }}_{k|k},\Theta_{k|k}$
**Initialization:**${\hat{x}}_{0,k|k}^{\left(0\right)}\leftarrow {\hat{x}}_{k|k-1},\;P_{0,k|k}^{\left(0\right)}\leftarrow P_{k|k-1},\;{\hat{\theta }}_{0,k|k}^{\left(0\right)}\leftarrow {\hat{\theta }}_{k|k-1},\;\Theta_{0,k|k}^{\left(0\right)}\leftarrow \Theta_{k|k-1},$$\alpha_{0,k|k}^{i,\left(0\right)}\leftarrow \alpha_{k|k-1}^{i},\beta_{0,k|k}^{i,\left(0\right)}\leftarrow \beta_{k|k-1}^{i}\mathrm{for}\;i=1,\ldots ,n_{y},$$y_{k}^{j,\left(0\right)}\leftarrow z_{k}^{j}\;\mathrm{for}\;j=1,\ldots n_{k},$$\Sigma_{k}^{y,\left(0\right)}\leftarrow E_{q_{X}^{\left(0\right)}}\left[\left(sX_{k}\right)\right]\;\mathrm{using}\left(30\right)$**Iterations:**
 **for** 
$n=0,\ldots ,n_{max}-1$
  Calculate $R({\hat{x}}_{n+1,k+1|k})$ using (9)
  **for** 
$\ell =0,\ldots ,\ell_{max}-1$
  Calculate $\left\{{\hat{x}}_{n+1,k|k}^{\left(\ell +1\right)},P_{n+1,k|k}^{\left(\ell +1\right)}\right\}$ using (24)  Calculate ${\left\{\alpha_{n+1,k|k}^{i,\left(\ell +1\right)},\beta_{n+1,k|k}^{i,\left(\ell +1\right)}\right\}}_{i=1}^{n_{y}}$ using (26)  Calculate ${\left\{{\hat{y}}_{n+1,k|k}^{\left(\ell +1\right)},\Sigma_{n+1,k|k}^{\left(\ell +1\right)}\right\}}_{i=1}^{n_{k}}$ using (29)  Calculate $\left\{{\hat{\theta }}_{n+1,k|k}^{\left(\ell +1\right)},\Theta_{n+1,k|k}^{\left(\ell +1\right)}\right\}$ using (32)
  **end for**
${\hat{x}}_{n+1,k|k-1}^{\left(0\right)}\leftarrow {\hat{x}}_{n,k|k}P_{n+1,k|k-1}^{\left(0\right)}\leftarrow P_{n,k|k},\;{\hat{\theta }}_{n+1,k|k-1}^{\left(0\right)}\leftarrow {\hat{\theta }}_{n,k|k},\;\Theta_{n+1,k|k-1}^{\left(0\right)}\leftarrow \Theta_{n,k|k},$$\alpha_{n+1,k|k-1}^{i,\left(0\right)}\leftarrow \alpha_{n,k|k}^{i},\;\beta_{n+1,k|k-1}^{i,\left(0\right)}\leftarrow \beta_{n,k|k}^{i}\;\mathrm{for}\;i=1,\ldots ,n_{y},$$y_{n+1,k|k}^{j,\left(0\right)}\leftarrow z_{k}^{j}\;\mathrm{for}\;j=1,\ldots n_{k},$   $\Sigma_{n+1,k}^{y,\left(0\right)}\leftarrow \Sigma_{k}^{y,\left(0\right)}$**end for**
where $\mu_{1},X_{1},\mu_{2},X_{2}$, respectively, represent the center position and range matrix of the two ellipses. The first term corresponds to the estimation error, and the second term corresponds to the range error. AGWD is the average value taken after multiple Monte Carlo simulation experiments. In addition, we define the RMSE calculation method taking position estimation as an example:(34)$${\mathrm{RMSE}}_{\mathrm{pos}}:=\sqrt{\frac{1}{M}\sum_{m=1}^{M}\left[{\left(x_{1,k}-{\hat{x}}_{1,k|k}^{m}\right)}^{2}+{\left(x_{2,k}-{\hat{x}}_{2,k|k}^{m}\right)}^{2}\right]}$$
where the superscript m represents the mth Monte Carlo operation, and $x_{i,k}$ represents the ith element of $x_{k}$. The error evaluation of the extension is calculated by the ratio of the estimated area to the target area. The average root mean square error (ARMSE) is simply the average of multiple RMSEs. It gives an overall measure of model error across different data points or experiments.

**Table 1 sensors-24-01362-t001:** The computational complexity of ETT-IDUCM iteration.

Formula	The Number of Flops
${\hat{x}}_{k|k-1}^{a}=F{\hat{x}}_{k-1|k-1}^{a}$	$2n^{2}-n$
$P_{k|k-1}^{a}=FP_{k-1|k-1}^{a}F^{T}+Q$	$4n^{3}-n^{2}$
$\alpha_{k|k-1}^{i}=\gamma \alpha_{k-1|k-1}^{i}$	m
$\beta_{k|k-1}^{i}=\gamma \beta_{k-1|k-1}^{i}$	m
${\hat{x}}_{n+1,k|k}^{\left(\ell +1\right)}=P_{n+1,k|k}^{\left(\ell +1\right)}(P_{n,k|k-1}^{-1}{\hat{x}}_{n,k|k-1}+n_{k}H_{k}^{T}{\tilde{\Lambda }}_{k}{\bar{y}}_{k}))$	$n^{3}+2n^{2}+2\mathrm{mn}$
$P_{n+1,k|k}^{\left(\ell +1\right)}={\left(P_{n,k|k-1}^{-1}+n_{k}H_{k}^{T}{\tilde{\Lambda }}_{k}H_{k}\right)}^{-1}$	$2n^{3}+2mn^{2}+n^{2}$
${\tilde{\Lambda }}_{k}:=\left(1-\exp\left(-2\Theta_{n,k|k}^{\left(\ell \right)}\right)\right)\frac{\mathrm{Tr}{\tilde{\Delta }}_{k}}{2}I_{2}+\exp\left(-2\Theta_{n,k|k}^{\left(\ell \right)}\right)\left(T_{{\hat{\theta }}_{n,k|k}^{\left(\ell \right)}}\tilde{\Delta }T_{{\hat{\theta }}_{n,k|k}^{\left(\ell \right)}}^{T}\right)$	$2m^{2}+2m$
${\tilde{\Delta }}_{k}:=\mathrm{diag}\left(\frac{\alpha_{n}^{1,\left(\ell \right)}}{s\beta_{n}^{1,\left(\ell \right)}},\frac{\alpha_{n}^{2,\left(\ell \right)}}{s\beta_{n}^{2,\left(\ell \right)}},\ldots ,\frac{\alpha_{n}^{n_{y},\left(\ell \right)}}{s\beta_{n}^{n_{y},\left(\ell \right)}}\right)$	2m
$\alpha_{n+1,k|k}^{i,\left(\ell +1\right)}=\alpha_{n,k|k-1}^{i}+0.5n_{k}$	m
$\beta_{n+1,k|k}^{i,\left(\ell +1\right)}=\beta_{n,k|k-1}^{i}+\frac{1}{2s}\sum_{j=1}^{n_{k}}{\tilde{\Psi }}_{k}$	m×(p+1)
${\tilde{\Psi }}_{k}:=E_{q_{\theta_{n}}^{\left(\ell \right)}}\left\{T_{\theta_{n,k}^{\left(\ell \right)}}^{T}\left[\left({\hat{y}}_{n,k}^{j,\left(\ell \right)}-H{\hat{x}}_{n,k|k}^{\left(\ell \right)}\right){\left({\hat{y}}_{n,k}^{j,(n)}-H{\hat{x}}_{n,k|k}^{\left(\ell \right)}\right)}^{T}+HP_{n,k|k}^{\left(\ell \right)}H_{k}^{T}+\Sigma_{n,k}^{y,\left(\ell \right)}\right]T_{\theta_{n,k}^{\left(\ell \right)}}\right\}$	$2m^{3}+2m^{2}n+2mn$
${\hat{y}}_{n+1,k}^{j,\left(\ell +1\right)}=\Sigma_{n+1,k}^{y,\left(\ell +1\right)}({\tilde{\Lambda }}_{k}H{\hat{x}}_{n,k|k}^{\left(\ell \right)}+{{\hat{R}}_{n+1}}^{-1}z_{k}^{j})$	$m^{3}-2mn^{2}+2m^{2}+m$
$\Sigma_{n+1,k}^{y,\left(\ell +1\right)}={\left({\tilde{\Lambda }}_{k}+{{\hat{R}}_{n+1}}^{-1}\right)}^{-1}$	$2m^{3}+m^{2}$
$\Sigma_{k}^{y,\left(0\right)}:=\mathrm{diag}\left(\frac{s\beta_{n}^{1,\left(0\right)}}{\left(\alpha_{n}^{1,\left(0\right)}-1\right)},\frac{s\beta_{n}^{2,\left(0\right)}}{\left(\alpha_{n}^{2,\left(0\right)}-1\right)},\ldots ,\frac{s\beta_{n}^{n_{y},\left(0\right)}}{\left(\alpha_{n}^{n_{y},\left(0\right)}-1\right)}\right)$	3m
${\hat{\theta }}_{n+1,k|k}^{\left(\ell +1\right)}=\Theta_{n+1,k|k}^{\left(\ell +1\right)}\left\{\Theta_{n,k|k-1}^{-1}{\hat{\theta }}_{n,k|k-1}+{\tilde{A}}_{k}\right\}$	3
$\Theta_{n+1,k|k}^{\left(\ell +1\right)}={\left\{\Theta_{n,k|k-1}^{-1}+{\tilde{B}}_{k}\right\}}^{-1}$	2
${\tilde{A}}_{k}:=\sum_{j=1}^{n_{k}}\mathrm{Tr}\left[{\tilde{\Delta }}_{k}{\left(T_{{\hat{\theta }}_{n,k|k}^{\left(\ell \right)}}^{\prime }\right)}^{T}{\tilde{\Omega }}_{k}\left(T_{{\hat{\theta }}_{n,k|k}^{\left(\ell \right)}}^{\prime }\right){\hat{\theta }}_{n,k|k}^{\left(\ell \right)}\right]-\mathrm{Tr}\left[\tilde{\Delta }T_{{\hat{\theta }}_{n,k|k}^{\ell }}^{T}{\tilde{\Omega }}_{k}\left(T_{{\hat{\theta }}_{n,k|k}^{\left(\ell \right)}}^{\prime }\right)\right]$	$(m^{3}+2m^{2}n)\times p$
${\tilde{B}}_{k}:=\sum_{j=1}^{n_{k}}\mathrm{Tr}\left[\tilde{\Delta }{\left(T_{{\hat{\theta }}_{n,k|k}^{\left(\ell \right)}}^{\prime }\right)}^{T}{\tilde{\Omega }}_{k}\left(T_{{\hat{\theta }}_{n,k|k}^{\left(\ell \right)}}^{\prime }\right)\right]$	$m^{3}\times p$
${\tilde{\Omega }}_{k}:=H_{k}P_{n,k|k}^{\left(\ell \right)}H_{k}^{T}+\Sigma_{n,k}^{y,\left(\ell \right)}+\left({\hat{y}}_{n,k}^{j,\left(\ell \right)}-H_{k}{\hat{x}}_{n,k|k}^{\left(\ell \right)}\right){\left({\hat{y}}_{n,k}^{j,\left(\ell \right)}-H_{k}{\hat{x}}_{n,k|k}^{\left(\ell \right)}\right)}^{T}$	$2m^{2}n$

Here, the dimensions of both $\theta_{k}$ and $\Theta_{k}$ are assumed to be one.

The extended target is an ellipse with semi-axes of $L_{a}^{*}=170m$ and $L_{b}^{*}=40m$. It starts moving from the origin at a constant speed of 50 km/h. The trajectory of this target is similar to the one in [6], consisting of a 45° and two 90° turns and straight paths. The measurements are generated by the scattering center through a uniform distribution, and the number of them follows a Poisson distribution, with parameter λ. The sampling period T = 10 s, the time decay constants of Fel-SCM is τ=50, the degree of freedom of Wen-UCM is δ=40, the scale variables of each algorithm are consistently s = 1/4. The process noise covariance of the motion state in [14] is diag[1,1,0.1,0.1,0.01], and the VB iteration $\ell_{max}$ is fixed to 5, while the number of IEKF iterations N is set to 4, the initial state $x_{0}=\left[100;-100;0;0;-pi/3\right]$, and the parameters of the other two algorithms are correspondingly adjusted to 4-dimensions. Figure 1 presents the measurements, trajectory, and estimation results from a single example run of several main comparison algorithms for $\sigma_{r}=50\;m$, $\sigma_{\varepsilon }=0.01\;\mathrm{rad}$, and λ=10.

We compared the root mean square error (RMSE) and average Gaussian–Wasserstein distance (GWD) over M=300 Monte Carlo runs. The results are shown in Figure 2 for $\sigma_{r}=50\;m$, $\sigma_{\varepsilon }=0.01\;\mathrm{rad}$ and λ=10. Table 2 lists the comparison of average running time for a single Monte Carlo simulation for each algorithm. Figure 3 shows the ARMSE of algorithm estimation due to changes in the major and minor axes of an elliptical extended target for $\sigma_{r}=50\;m$, $\sigma_{\varepsilon }=0.01\;\mathrm{rad}$, λ=20, and $L_{a}=L_{a}^{*}/k,L_{b}=L_{b}^{*}\cdot k$. Figure 4, Figure 5 and Figure 6 compare ARMSE by changing only one parameter.

Given different extended target sizes and dynamic error parameters, as shown in Figure 2, Figure 3, Figure 4, Figure 5 and Figure 6, the ETT-IDUCM consistently outperforms the other three algorithms, particularly when the extended target is executing a turn. This demonstrates that the method proposed in this paper is more effective compared to the other methods. From Figure 2, it is observable that all algorithms experience fluctuations in error when the target undergoes turning movements, but the timing of the peak values differs. This is because when the target performs weak maneuvering, other algorithms, due to poor handling of nonlinearity, have a bias in extension estimation, which becomes more accurate gradually, through the accumulation of measurements. In contrast, our proposed ETT-IDUCM algorithm can accurately estimate the extension from the beginning, thus showing stronger adaptability after the maneuver ends, and this can be intuitively seen in Figure 1. From Figure 4, it can be seen that, as the measurement error increases, the error growth for Tun-UCM and ETT-IDUCM accelerates, compared to the other two algorithms. This acceleration occurs because, with larger measurement errors, the measurement point set fails to accurately depict the target’s extension. The two models, which are more finely modeled, experience a rapid increase in algorithm error under such conditions. On the other hand, the other two algorithms, despite their less precise modeling, exhibit a slower divergence as errors increase, leading to a slower change in their adaptability.

The update process of Tun-UCM uses only VB technology for derivation, which leads to a significant performance reduction when the sensor error increases, and the filter tends to diverge. ETT-IDUCM not only improves the correlation of the covariance after conversion, but also combines DUCM with the iterative process of the IEKF to continuously obtain more accurate state predictions and related covariances. This covariance update will further affect the extension estimation, which is directly reflected in the target turning stage.

### 4.2. Comparison of Update Iterations of ETT-DUCM

Furthermore, set $\sigma_{r}^{*}=10\;m$, $\sigma_{\varepsilon }^{*}=0.002\;\mathrm{rad}$, λ=20, based on 100 Monte Carlo simulations. Figure 7 shows the ARMSE simulation result curves of the ETT-DUCM algorithm under different iteration numbers N, when the measurement errors increase synchronously.

As shown in Figure 7, with the increase of iteration times N, the position and speed errors in ETT-IDUCM are significantly reduced, and the average GWD also changes accordingly. From the subplots in Figure 7c, the comparison of results at different numbers of iterations N with N=0 shows that the estimation accuracy of the extension improves with the number of IEKF iterations. Due to the presence of measurement errors, these points cannot accurately reflect the evolution of the target’s extension. Furthermore, the strong nonlinearity of the contour means that improvements in extension estimation are not as marked as those seen in centroid state estimation. Additionally, considering the trade-off between running time and filtering precision, setting the value of N to 4 is found to be a practical choice. As the level of measurement error increases, the robustness of the algorithm is demonstrated.

## 5. Conclusions

A decorrelated unbiased conversion method is used to handle nonlinear measurements in polar coordinates in extended target tracking with random matrices, to fully consider the impact of measurement noise and estimation bias. Moreover, this paper develops a method that combines iterated extended Kalman filter updates with variational Bayesian cycles, which cannot only accurately estimate the target’s kinematic state and extension but also provide more and more accurate parameters for DUCM to better and better deal with nonlinear measurements, from polar coordinates to Cartesian coordinates. As the number of iterations increases, the estimation accuracy is significantly improved.

Simulation results show that the estimation accuracy of the proposed method is superior to that of the existing ETT methods when using nonlinear measurements, especially when the target is performing weak maneuvering movements, in terms of different levels of measurement noises and numbers of measurements.

In future work, we will focus on tracking algorithms for extended targets performing aggressive maneuvering movements. Considering the current algorithm’s performance, integrating it with an interactive multiple model (IMM) approach may be worth exploring. This potential combination could offer a more robust solution for accurately tracking extended targets under highly dynamic conditions, thereby enhancing the adaptability and effectiveness of the tracking system in complex scenarios.

## Figures and Tables

**Figure 1 sensors-24-01362-f001:**
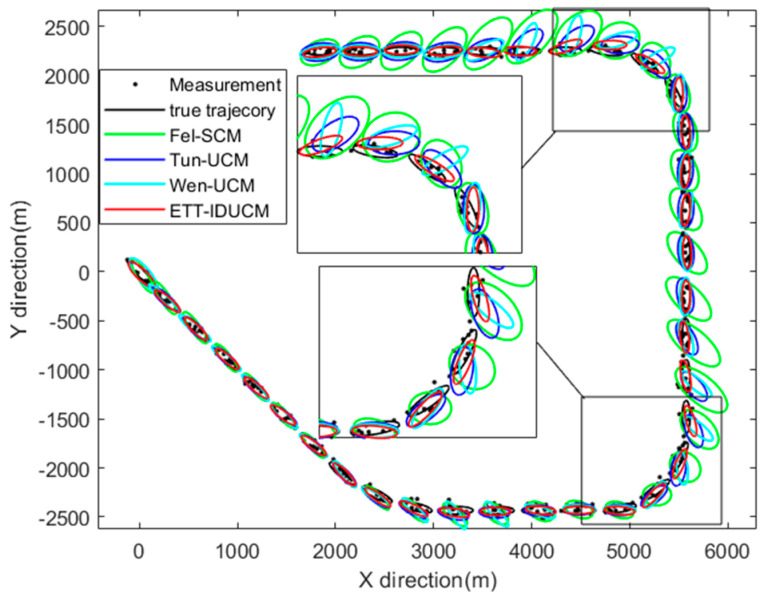
Measurements, trajectory, and estimation results of a single example run.

**Figure 2 sensors-24-01362-f002:**
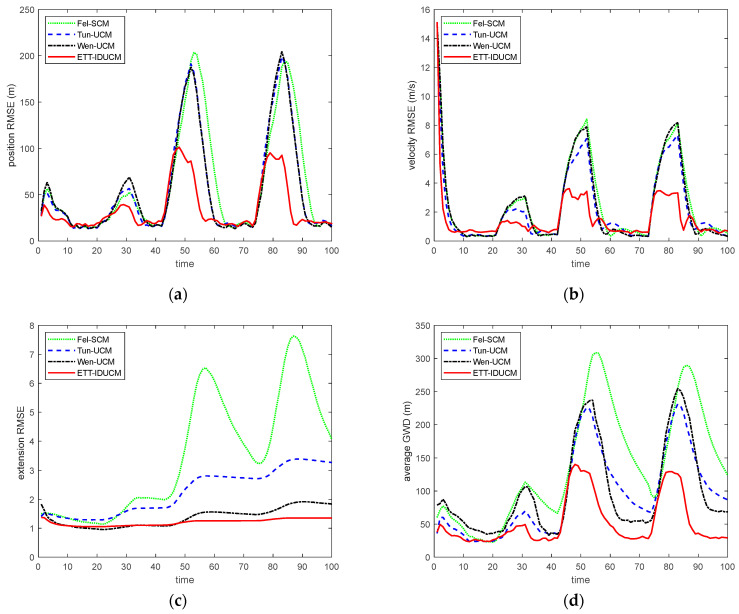
RMSE and average GWD with $\sigma_{r}=50\;m$, $\sigma_{\varepsilon }=0.01\;\mathrm{rad}$, λ=10. (**a**) RMSE of position; (**b**) RMSE of velocity; (**c**) RMSE of extension; (**d**) average GWD.

**Figure 3 sensors-24-01362-f003:**
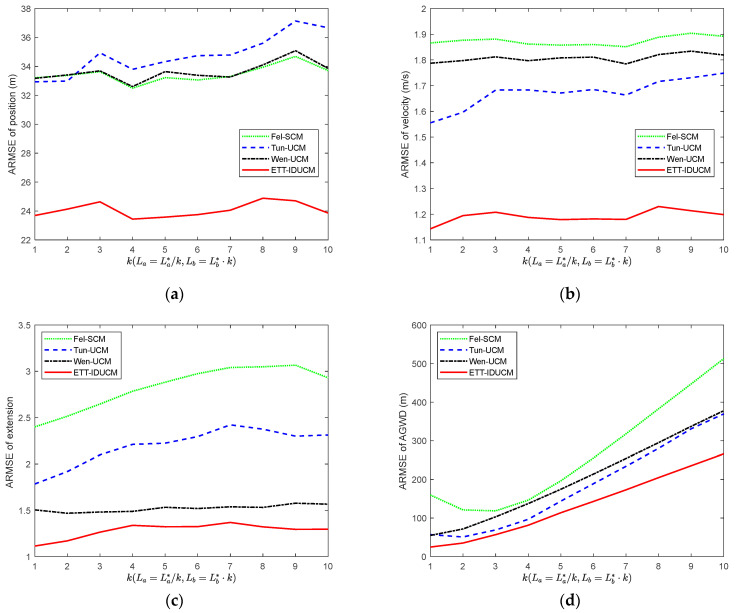
RMSE and average GWD with $\sigma_{r}=50\;m$, $\sigma_{\varepsilon }=0.01\;\mathrm{rad}$, λ=20, $L_{a}=L_{a}^{*}/k,L_{b}=L_{b}^{*}\cdot k$. (**a**) RMSE of position; (**b**) RMSE of velocity; (**c**) RMSE of extension; (**d**) average GWD.

**Figure 4 sensors-24-01362-f004:**
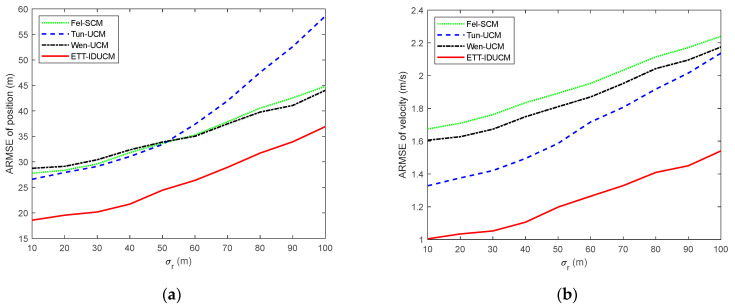
ARMSE with $\sigma_{r}=10\;m–100\;m$, $\sigma_{\varepsilon }=0.01\;\mathrm{rad}$, λ=20. (**a**) ARMSE of position; (**b**) ARMSE of velocity; (**c**) ARMSE of extension; (**d**) ARMSE of AGWD.

**Figure 5 sensors-24-01362-f005:**
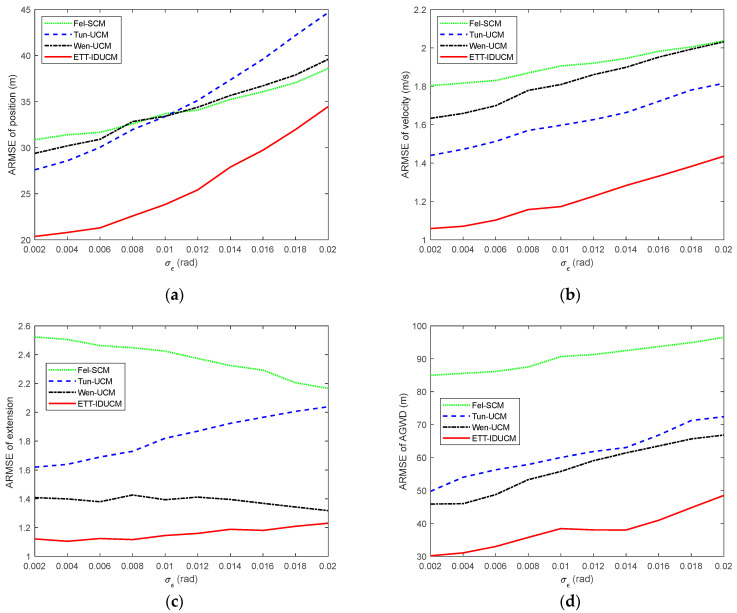
ARMSE with $\sigma_{r}=50\;m$, $\sigma_{\varepsilon }=0.002\;\mathrm{rad}–0.02\;\mathrm{rad}$, λ=20. (**a**) ARMSE of position; (**b**) ARMSE of velocity; (**c**) ARMSE of extension; (**d**) ARMSE of AGWD.

**Figure 6 sensors-24-01362-f006:**
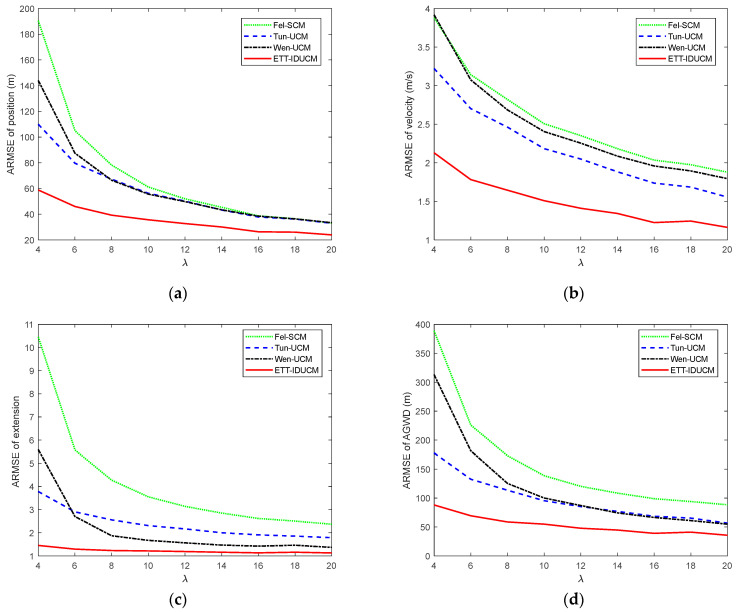
ARMSE with $\sigma_{r}=50\;m$, $\sigma_{\varepsilon }=0.001\;\mathrm{rad}$, λ=4–20. (**a**) ARMSE of position; (**b**) ARMSE of velocity; (**c**) ARMSE of extension; (**d**) ARMSE of AGWD.

**Figure 7 sensors-24-01362-f007:**
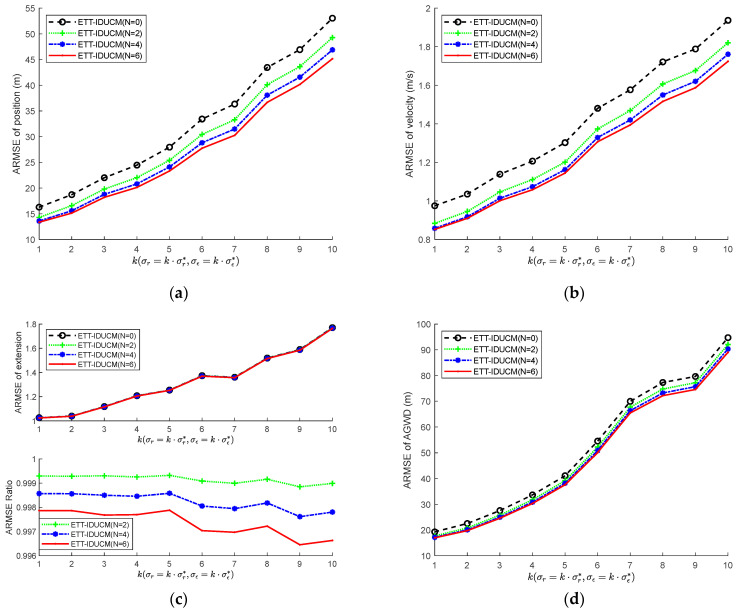
ARMSE with $\sigma_{r}=k\cdot \sigma_{r}^{*}$, $\sigma_{\varepsilon }=k\cdot \sigma_{\varepsilon }^{*}$, λ=20. (**a**) ARMSE of position; (**b**) ARMSE of velocity; (**c**) ARMSE of extension; (**d**) ARMSE of AGWD.

**Table 2 sensors-24-01362-t002:** Average running time for each algorithm.

Algorithm	Time/ms
Fel-SCM	173.918
Tun-UCM	84.971
Wen-UCM	194.494
ETT-IDUCM	270.697

## Data Availability

The data that support the findings of this study are available from the corresponding author upon reasonable request.

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
