# Peer review of "An Iteratively Extended Target Tracking by Using Decorrelated Unbiased Conversion of Nonlinear Measurements"

_sensors, 2024, doi:10.3390/s24051362_

Round 1

Reviewer 1 Report

Comments and Suggestions for Authors

In this paper, an approach is proposed for Extended Target Tracking (ETT) based on random matrices, addressing the challenge of nonlinear measurements in practical tracking applications. An iteratively extended target tracking method (ETT-IDUCM) is derived, which employs a Decorrelated Unbiased Converted Measurement (DUCM) method and combines Iteratively Extended Kalman Filter (IEKF) updates with Variational Bayesian (VB). This proposed approach not only improves the estimation of the target's kinematic state and extension but also enhances the handling of nonlinear measurements from polar coordinates to Cartesian coordinates. Simulation results show that the proposed ETT-IDUCM algorithm outperforms the comparable methods in estimating the target's kinematic state and extension. The reviewer’s major comments are as follows.

1. The reviews of related work are not inadequate, and many related methods are missing, such as some works about adaptive and robust Kalman filtering based on the Variational Bayesian method.

2. The theoretical analysis is incomplete. This paper proposes to use IEKF to track the target, which improves its accuracy but also increases the computational cost compared to using KF. The author should briefly analyze the computational cost of this method in this paper and compare it with other existing methods in terms of computational cost in the simulation.

3. The simulation analyses are also insufficient. The estimation accuracy of the extension does not improve with the increase of the number of iterations in the simulation. The author should explain the reasons for this phenomenon in the simulation.

4. Formula (11) is incorrect. Formula (11) should be the conditional probability density of y_k^i, not the conditional probability density of z_k^i. The author should revise formula (11) and check this paper carefully for other writing or formula errors.

5.The description of the external IEKF loop and the internal VB loop in Section 3.2 could be more detailed. Please provide necessary explanations for the derivation process of Equation 32.

6. To clearly illustrate the performance of each algorithm, please include measurement points and estimations from other algorithms in Figure 1.

7. Please provide additional explanation for the error fluctuations that appear in Figure 2.

8. There are some grammatical errors, for example, the sentence in the first paragraph, "Consequently, a target might be considered an extended target (ET)..." should be corrected as "Consequently, a target might be considered as an extended target (ET)...".

Comments on the Quality of English Language

none

Reviewer 2 Report

Comments and Suggestions for Authors

The number of references is too many and it should be reduced. The main results should be compared  by the results given in references [8] and [26]. 

Reviewer 3 Report

Comments and Suggestions for Authors

This paper introduces an innovative approach, ETT-IDUCM, addressing the challenge of nonlinear measurements in extended target tracking based on random matrices. By employing Decorrelated Unbiased Converted Measurement (DUCM), it converts nonlinear measurements, such as range and azimuth, into Cartesian coordinates, mitigating issues related to unknown scattering centers. The method combines Iteratively Extended Kalman Filter (IEKF) updates with Variational Bayesian (VB) cycles for precise estimation of the target's kinematic state and extension. The external IEKF iteration and internal VB iteration contribute to refining predictions and enhancing measurement noise covariance. The simulation results affirm the algorithm's superiority, demonstrating its enhanced precision in estimating both the target's kinematic state and extension compared to existing methods. ETT-IDUCM presents a promising advancement in extended target tracking for practical applications. So, I recommend this paper for publication.

Reviewer 4 Report

Comments and Suggestions for Authors

Dear Author,

1. State-dependent statistics need to elaborate more on statistical data and how it influences target kinematics.

2. Equation 3: Elbarate in more detail in the annexure, similarly to other equations.

3. Kinamatic state line number 259, elaborate more details with references.

4. Comparison of the tracking accuracy of different algorithms compared with other modules

5. Figuer 2, ETT SCM, UCM data are showing a peak; can you discuss more technically why slants to words take more time?

6. Why are ETT SCM, ETT-UCM cross-linking at 90 in figuer 3(d) 

7. how novel your work with other articles.

8. Justify why and how it is influence with this model and analysis weak maneuvering movements in this work

Round 2

Reviewer 2 Report

Comments and Suggestions for Authors

The authors followed the comments and corrected the paper.

Comments on the Quality of English Language

It needs minor revision.